# Will Genomic Information Facilitate Forest Tree Breeding for Disease and Pest Resistance?

Richard A. Sniezko [1,*] , Jennifer Koch [2], Jun-Jun Liu [3] and Jeanne Romero-Severson [4]

1 USDA Forest Service, Dorena Genetic Resource Center, 34963 Shoreview Drive, Cottage Grove, OR 97424, USA
2 USDA Forest Service, Northern Research Station, Delaware, OH 43015, USA; jennifer.koch@usda.gov
3 Canadian Forest Service, Natural Resources Canada, Victoria, BC V8Z 1M5, Canada; jun-jun.liu@nrcan-rncan.gc.ca
4 Department of Biological Sciences, University of Notre Dame, Notre Dame, IN 46556, USA; jeanne.romero-severson.1@nd.edu
* Correspondence: richard.sniezko@usda.gov

**Abstract:** Forest trees are beleaguered by the ever-increasing onslaught of invasive pests and pathogens, with some species in danger of functional extinction. Recent successes in developing resistant populations using traditional tree breeding assures that some of the affected species will persist in future forests. However, the sheer number of threatened species requires increases in breeding efficiency. The time is right to consider how the use of genomic resources might aid breeding efforts in the next 20 years. Any operational benefit of genomic resources will be minimal without closer collaboration between tree breeders, forest managers, and genomic researchers. We reflect here on what attributes were responsible for the success of traditional resistance breeding programs and whether advances in genomics can realistically accelerate breeding. We conclude that the use of genomics to directly advance resistance breeding efforts in the next 20 years will be limited. Major obstacles will include factors such as the undomesticated nature of most tree species, the quantitative genetic nature of resistance in many species, and the lack of adequate funding to accelerate and more fully develop genomic resources. Despite these limitations, genomic tools have potential to help increase our understanding of the nature of resistance, and the genetic variability in the host, which can aid in the deployment of resistant populations and may assist in marker-assisted selection, particularly for major gene resistance.

**Keywords:** biotechnology; disease and pest resistance; forest trees; genomic resources; resistance breeding; restoration; tree improvement

## 1. Introduction

The rapid spread of invasive forest tree pests and pathogens has resulted in the extensive mortality of forest tree species throughout the world [1–3]. In many cases, management activities, including the introduction of biological control agents, have had a limited impact on tree mortality caused by the biotic invaders. More focused efforts to develop biocontrol agents have been suggested for a variety of disease management approaches [4], but some level of host resistance will likely still be needed to insure sustainable populations. By contrast, traditional tree breeding programs, notably those utilizing the genetic variation within a species (as opposed to inter-crossing species to impart resistance or using genetic engineering or gene editing) have had notable successes [5], encouraging recent calls for even more concerted efforts as well as the use of genomic resources to aid resistance breeding efforts [6–9].

Unfortunately, the development of extensive genomic resources for tree species has not contributed directly to the success of resistance breeding programs. An examination of this irony reveals that the developers of genomic resources in forest trees often

act independently of operational breeding or even resistance breeding research and forest genetics research. In contrast, plant breeders working on row crops have achieved considerable success integrating genomic resources into operational breeding to produce disease-resistant, higher-yielding, higher-quality products [10]. However, unlike many crops, trees are perennial and will need to survive, often in unmanaged forests, for decades to centuries. Most forest tree species are undomesticated, with only a few involved in active breeding programs for any trait. The lack of extensive knowledge on the basic biology and genetics, the large genome size (notably of conifers), and the modest funding available for most tree species also limits their application. The number of years before many tree species begin to flower also is a constraint.

It is important to recognize that tree breeding programs focused on restoration of resistant trees into natural populations have different goals and are more complex than row crop resistance breeding programs. For tree species heavily impacted by pathogens or pests, the objective may be the development of adapted populations with enough genetic diversity and durable resistance to ensure the persistence of the species, rather than the development of cultivars selected for performance in plantation or orchard environments. The native range of many forest tree species is divided into seed zones to ensure that planted stock is descended from source populations adapted to the environments in which their descendants are planted [11]. For forest restoration, there may be 30 or many more resistant parents used in a seed orchard for each seed zone. Achieving these target goals is a massive undertaking. This perspective will examine to what extent the early integration of genomic resources within traditional breeding programs might provide supportive research and, in some cases, may have the potential to shorten development time while, at the same time, improve the quality of the results.

## 2. Framing the Problem

Resistance breeding programs for trees have existed for more than 60 years. Once the damaging agent is identified and the host symptoms are characterized, the initial objective is to discover the extent and degree of resistance in native populations. In host tree species in which resistance has a simple genetic basis, large-scale screening enables the identification of additional resistant trees, the development of breeding programs, and the selection of elite parents and progeny for the establishment of seed orchards for each seed zone [12]. This approach, if properly implemented, works well, but can require testing the progenies of thousands of parent trees, a time-consuming and costly process that can take decades [13]. Most resistance programs spend a significant amount of time and resources developing methods to distinguish resistant from susceptible phenotypes so that candidate resistant parent trees initially selected from natural stands can be validated. However, many of these candidates are found to have little or no resistance. With the increased need and interest in resistance breeding, a methodology to identify heritable resistance more effectively in natural stands would be a game changer for resistance breeding programs. In the cases where it may be feasible, it could potentially greatly decrease the number of field selections to use in inoculation trials, shortening both the time and cost involved in the vital first generation. Thus, this is a key arena that genomic information might have the capability of assisting with in the future.

Genomic tools have and continue to be successfully utilized in the development and/or operation of resistance breeding programs for specific purposes that include the assessment of the genetic diversity of breeding populations, confirmation and tracking of parentage and relatedness, and evaluation of pollen contamination/flow in seed orchards. Despite these successes, in truth, the relationship between genotype and phenotype is rarely simple. The detection of genomic variation in the responses of the host tree species to the invasive pest or pathogen requires a reproducible method of distinguishing the phenotypes that range from fully susceptible to highly resistant in segregating populations (cloned genotypes and full- or half-sibling families) to ascertain the heritability of resistance

phenotypes. Phenotyped segregating populations are necessary to even begin to identify potentially informative genetic markers for the desired traits in the host species of interest.

In the cases of major gene resistance (MGR) conveyed by a single dominant gene for complete resistance with a bimodal phenotypic segregation in progeny, the identification of genetic markers is likely the simplest. By contrast, quantitative resistance (QR) shows a continuous distribution of phenotypes from resistance to susceptibility in a progeny under the influence of many quantitative trait loci (QTLs) with small effects [14,15]. Each QR-related QTL may contain several to hundreds of functional genes with complex environmental and/or developmental interactions. This complexity with less-known molecular mechanisms hinders the prospect of resistant QTLs being fully useable in the near future. In theory, genomic resources could enable the characterization of the genetic mechanisms that result in resistance responses, at least in the cases of MGR and QTLs with stable major effects, allowing tree breeders to identify some of the resistance in natural populations without labor-intensive screening.

The failure of this often-promised result to materialize is due to the reality that this level of gene discovery and characterization requires the sustained use of genomic resources as well as the tree breeding infrastructure (field plots, greenhouses, wet labs, and support personnel) and genetic resources (phenotyped families of known pedigree) required for traditional breeding. This integration of genomics, genetics, and the infrastructure needed for both requires substantial and sustained financial support. Often, one or more of these prerequisites is missing, with sustained financial support being the most notable.

The mass destruction caused by emerald ash borer (EAB, *Agrilus planipennis* Fairmaire) has led to the listing of five North American ash species (*Fraxinus* spp.) on the International Union for Conservation of Nature (IUCN) red list as being in the highest risk category for extinction (IUCN.org). The discovery of EAB in the United States in 2002 provides an example of the impact that the lack of genetic resources can have when a highly damaging new invasive pest is introduced. Research that proposed to identify the mechanisms or markers of EAB resistance reported from 2004 to 2015 focused on using 'omics' techniques to compare the constitutive (trees not infested with EAB) expression levels of genes (transcriptomics), proteins, and metabolites between a commercially available cultivar of an EAB-resistant ash species and various susceptible species, each represented by either a single cultivar (or a single seed lot). Genes, proteins, or phenolic compounds that were present at higher levels in the resistant cultivar relative to the susceptible cultivars were identified as having potential involvement in resistance with the hope that they could be developed into useful biomarkers for resistance in support of breeding programs that did not yet exist [5]. The functional roles of these candidate resistance compounds, genes, and proteins have still not been validated. Given that the experimental designs used did not account for interspecific and intraspecific genetic differences unrelated to EAB defense responses and the imprecise phenotyping employed, the results of these investigations have limited practical utility. Wise investments in the development of appropriate genetic resources (phenotyped populations of known pedigree) can provide the foundation necessary for the subsequent integration of genomics, increasing the chances of success. Methods to screen for EAB resistance have now been developed (without the use of genomics), providing evidence that select trees ("lingering ash") that have survived in long-infested natural forests have increased defense responses against EAB [16]. Recently, it has been reported that the seedling families of lingering green ash (*F. pennsylvanica* Marsh.) parents mount induced defenses against EAB and are more resistant to EAB than the families of unselected trees [17].

### 2.1. Four Limiting Factors

There are many factors that have limited the utility of genomics in resistance breeding in trees, even when the infrastructure required for traditional resistance breeding exists. The first major factor includes financial limitations. Although there are some tree species of relatively high economic importance used in plantation forestry, there are many more

threatened species of primary importance for ecosystem services other than for wood or pulp production. In general, the undomesticated species of major ecological but minor economic importance have not commanded the economic resources necessary to sustain long-term resistance breeding programs. Most tree breeding programs, regardless of the economic importance of the species, lack the sustained financial support that has enabled major crop plant programs to make remarkable gains from selection using both traditional breeding and the successful integration of genomic technologies with traditional breeding.

The second factor is biological: the fundamental difficulty of defining resistance phenotypes in long-lived perennial species. Tree breeding programs have achieved success by testing segregating and clonal populations. However, early resistance assessments on such material still must be monitored over many years to determine the durability of resistance. In general, any utility of genomic resources will require the strong documentation of resistant and susceptible phenotypes, and this can take a decade or more.

The third factor is the difficulty of maintaining genetic diversity within the breeding population. Discovering a single genotype with resistance may be adequate in some crop species, but it is inadequate for forest trees. Long-lived forest tree populations must retain enough genetic diversity to buffer against the abiotic and biotic challenges that will arise in heterogenous ecosystems across timespans of hundreds of years, permitting a species to evolve under the pressures of the changing environment. In addition, multiple breeding populations are required to capture adaptive genetic capacity across the geographic range of most species.

The fourth factor is the complexity of the genetics that confer resistant phenotypes. Resistance can be conferred by a dominant resistance (R) allele for MGR as well as by QTLs for QR. QR is generally more desirable for durable resistance in long-lived forest trees, as multigenic mechanisms are less likely to be overcome by rapidly evolving pests and pathogens [14]. Working with QR is more challenging than working with MGR but gain from selection for quantitative disease resistance traits is achievable [18–21]. In many cases, at least one additional cycle of breeding will be needed to enhance the level of QR. In at least some forest tree species, any breeding is delayed a decade or more until the trees become reproductively mature.

In the next section, we draw on our multi-decade experience with applied resistance breeding programs to illustrate the extent to which the genomic information has contributed to successful outcomes or might do so in the next one or two decades. As examples, we discuss programs breeding for resistance to *Cronartium ribicola* J.C. Fisch. (the fungal pathogen that is the cause of white pine blister rust, WPBR) and offer insight into the realistic uses of genomics technologies to assist breeding programs. We also very briefly discuss the more recent program to develop resistance to emerald ash borer in some species of *Fraxinus*.

### 2.2. The Deceptive Simplicity of MGR

*Cronartium ribicola*, the fungal pathogen of WPBR, was introduced in the United Sates in the early 1900s. Mortality from WPBR infection can exceed 95 percent in some populations. The high economic value of three susceptible species, western white pine (*Pinus monticola* Dougl. ex D. Don), sugar pine (*P. lambertiana* Dougl.), and eastern white pine (*P. strobus* L.) provided the justification for funding investigations to find genetic resistance within these species and to develop management methods to mitigate disease spread and impact. The result was funding for the WPBR resistance breeding programs more than 60 years ago that continue today. The regional programs for *P. monticola*, now regarded as a major success for disease management, resulted from the willingness to make a sustained investment in the research and development necessary to prevent the loss of an economically valuable five-needle pine species. This investment enabled collaborations between the private and public sectors and across scientific disciplines that have resulted in success in resistance breeding using traditional plant breeding and improved management approaches for these species as well as for several other conifers affected by pathogens [5,12].

The complexity of this pathosystem began to reveal itself as scientists were able to apply genomic tools to the breeding populations of trees of known phenotypes, pedigree, and provenance produced through the ongoing rust-resistance breeding programs. Environmental influences on the phenotype, genetic variation in the pathogen, long breeding cycles, and generally low level of quantitative resistance in some of the five-needle pines [13] pose challenges in breeding for resistance to *C. ribicola*. MGR is documented in four of the nine five-needle pine species native to the U.S. [13,22,23], but in three of these species, the R genes map to different linkage groups and R-linked DNA markers and R-positional candidate polymorphisms have been developed to potentially exploit marker-assisted-selection (MAS) in these species in the future [24].

MGR may be controlled by the same orthologous locus in limber pine (*P. flexilis* James) and southwestern white pine (*P. strobiformis* Engelm.) [25]. However, recent preliminary evidence from genomic studies suggests there might be multiple alleles for a MGR locus, or a second R gene for MGR in limber pine (Liu and Sniezko, unpublished). In other pathosystems, such as *Phytophthora lateralis* Tucker & Milbrath resistance in Port-Orford-cedar (*Chamaecyparis lawsoniana* [A. Murr.]), genomic information, when available, will also be of use in discerning whether there is more than one locus for MGR present [20]. In the southern pines, genomics has led to the discovery of more than one MGR locus in loblolly pine (*P. taeda* L.) for resistance to fusiform rust, which is caused by the fungal pathogen *Cronartium quercuum* (Berk.) Miyabe ex Shiraif. sp. *fusiforme* [26,27]. Despite the intensive study of rust pathosystems in pines, no R gene has been cloned and functionally verified in any conifer species. Further work will be needed to fully develop the potential to use MAS for MGR for most tree species. The most effective tools for the prediction, selection, and comparison of MGR loci are developed from genomic sequences of R genes themselves. As elite resistant germplasm and genomic resources (such as genome and transcriptome sequences) are available from various forest breeding programs, the targeted re-resequencing (such as SMRT-RenSeq and amplicon-seq) of the plant R gene families of forest populations will help categorize novel R genes for MGR to different pathogens/pests in forest tree species [28,29].

### 2.3. The Challenges of QR Detection and Application

The strategic challenge of traditional breeding for QR against highly destructive, invasive pests and diseases is accomplishing the goal in the shortest amount of time. Rapid invasions can inflict irreparable damage in plantations and natural stands during the time it takes to complete one traditional breeding cycle. The question is not whether gain from selection for quantitative traits is possible in trees [20,21,30], the question is how feasible this undertaking is, at the pace and scale necessary, in the many species of need, when starting with undomesticated populations. In these species, under current practices, the cost and logistics of the testing of thousands of initial phenotypic selections to find the relatively few that are resistant for a species can take 20 years or more. Although MAS tools are successful in some cases for QR selection and pyramiding in crop breeding, the accurate detection of resistant QTLs for marker development is still not feasible in trees due to the long time period needed for generating mapping populations using controlled crosses. The role of genomics in QR breeding in trees was, until recently, limited to gene discovery and the verification of gene discovery in well-characterized pathosystems. One of the best illustrations of this is, again, the WPBR pathosystem.

Populations with QR against WPBR are well-documented in several tree species [21,30], some with major phenotypic effects [23]. The genetic mapping of QTLs associated with resistance usually requires $F_1$ mapping populations from controlled crosses between heterozygous parents, with the replication of mapping populations to permit phenotyping in different environments [31]. However, each cross can only contain the few QTLs present within a closed pedigree [32]. In contrast, GWASs have the potential to detect many QTLs in random populations without known family structure, but requires large population sizes, high-density genome-wide markers to detect alleles with small phenotypic effect,

and the high-quality phenotyping of every individual. The integration of GWAS, QTL, and linkage mapping has revealed a set of sugar pine genes potentially involved in both QR and MGR to WPBR [33]. However, the phenotypic effects and functions of these QR candidate genes are still uncertain in related seed families. Compared to QTLs and QR candidate genes with small effects, the search for QR families with major phenotypic effects and the confirmation of their inheritance will be more feasible. Financial support, rapid and highly effective phenotyping, ease of clonal propagation, and established infrastructure for screening and propagation has enabled the effective use of genomics for the initial phase of R gene discovery and QTL dissection in the WPBR pathosystems. However, a much more concerted effort is needed to fully incorporate genomic information for QR screening into operational forest tree resistance breeding. Even with such an effort, it may be primarily limited to the few QTLs of major effect and not a replacement for capturing a fuller array of the many genes involved in QR.

### 2.4. Potential Benefits of Comparative Genomics

Resistance to WPBR has been documented in eight of the five-needle white pine species native to the U.S. [13]. No information is yet available comparing the genomic basis of resistance in the North American species to those native to Eurasia, where there has been coevolution of the pine species and the pathogen, and where the pathogen has generally had little impact on the pines [34] which have shown high levels of resistance in screening trials [35]. A future study of the co-linearity and functional synteny of the plant R gene families between North American and Eurasian white pines could potentially be used to predict the functions of R genes for the evolutionary adaptation of genetic resistance to WPBR and develop MAS tools in related species. More R genes may be present in Eurasian white pines, native hosts to *C. ribicola*, but little information is available on the underlying nature of resistance in these species. Although inheritable resistance to WPBR was successfully introgressed from Himalayan white pine (*P. wallichiana* A.B. Jackson) into eastern white pine by interspecific hybridization [36,37], specific R genes are not characterized in *P. wallichiana* and its interspecific hybrids. The R-gene markers to guide introgression or to pyramid QTLs in specific pedigrees is one of the most realistic uses of MAS tools, but few programs are likely to pursue introgression with related species. High fidelity (HiFi) long-read sequencing enables the accurate comparison of the complex genomes of different forest species or the different families of one species, enabling the design of cloning strategies for specific rare R alleles with the long-term goal of the MAS-guided incorporation of broad resistance to a wide spectrum of pathogenic races of rust fungi [38,39].

Whole genome sequences are currently lacking for most tree species. As the cost of high-quality whole genome sequencing falls, more tree species and, notably, the conifers, which have very large genomes, will be sequenced. The availability of a reference genome sequence and high-density genotyping array enables the comparison of the genomic regions of captured alleles between different populations with known genome sequences from other studies, especially those from populations in disparate geographical regions or seed zones [40]. The characterization of family-specific candidate alleles for MGR and QR is promising to accelerate the breeding of loblolly pine for resistance to native fusiform rust [27]. Extending the utility of genomics even further, the comparison of sequenced MGR alleles in species challenged with a non-native invasive pest or pathogen to similar sequences in species where coevolution has occurred may provide some insights on the potential for durable resistance or indicate whether existing resistance might be enhanced by the introgression of MGR or QR resources from native to non-native host tree species.

An initial investigation of the EAB-*Fraxinus* pathosystem, in which no MGR alleles were identified, compared the sequences of over 1400 orthologue groups found across 26 taxa of the genus *Fraxinus* to look for evolutionary patterns of variation that co-occur in EAB-resistant species. This approach was based on the hypothesis that molecular convergence may be associated with convergent phenotypes [41] and has been used to successfully

identify sequence convergence among hearing genes in echolocating mammals [42,43]. Using this novel approach, 53 candidate genes were identified, many of which were homologues to genes known to be involved in insect resistance in other plants [44]. Comparative studies like this may be of limited value without knowledge of both sequence variation and phenotypic variation in response to EAB within species as well as between species. To be of any use to EAB resistance breeding programs, the functional role of these or any other proposed candidate genes will first have to be confirmed, which in and of itself will be a costly and long-term effort. It may be better to direct efforts toward the characterization of the EAB pathosystem in a few related species, using progeny tests and detailed phenotyping. Once validated, markers for candidate genes or other genomic features identified in this well-characterized system can be used to guide introgression through hybridization, if susceptible and resistant species are genetically compatible. Unfortunately, species in the Meliodes section of *Fraxinus* genus, which includes green ash and white ash, are genetically incompatible with the EAB resistant Asian *Fraxinus*. However, once functional and genetic studies have validated candidate genes or genomic features, sequence comparison within and between species may uncover novel candidate EAB resistance-associated variants in susceptible species. The individual genotypes of a susceptible species that are found to have either a resistance-associated gene variant or a novel variant, or combinations of variants, may be prioritized for EAB resistance phenotyping and for use in breeding to pyramid additional variants. Approaches like this require years of field testing and readily available, affordable genomics tools for screening hundreds and even thousands of samples. The limited funding currently available for functional studies on the EAB-*Fraxinus* pathosystem will slow efforts to identify and validate candidate genes and other genomic features, diminishing the likelihood of such efforts expediting the breeding process in the near term.

### 2.5. Operational Breeding for Resistance: Beyond Gene Discovery

Genetic mapping, GWAS, and genome sequencing will remain academic until the technology provides methods to rapidly identify superior parents and progeny, enhance gain from selection, maintain genetic diversity, and shorten the time to delivery (the time when trees with durable resistance are available for plantations or restoration). High throughput phenomics technology can permit rapid screening for QR resistance in parents and progeny, provided that the pathosystem has a phenotype visible to the instruments, but further development will be needed for application to forest tree species [45].

Breeding by Design (BBD) [46], the full integration of genomics technology and breeding strategies to accomplish the incorporation of QTLs for major and minor resistance to biotic and abiotic stress while maintaining productivity, is now a reality in rice breeding [47]. The success of BBD in rice and other row crops shows that the approach is feasible when the basic criteria are met. BBD strategies require considerable upfront resources: an array of structured, pedigree-verified populations or a very large set of randomly mating unstructured populations, a thoroughly vetted method of phenotyping and a set of high-density genetic markers informative for the species or genus of interest. The development of these necessary resources in a resistance breeding program in trees may take at least 10 years or more of concerted effort, which will only be possible with sustained funding. This will be not generally be feasible for most tree species, particularly those lacking high commercial value.

In some systems, such as with resistance to WPBR in whitebark pine (*P. albicaulis* Engelm.) and resistance to beech bark disease in American beech (*Fagus grandifolia* Ehrh.) [48], the first generation of selection may yield resistance high enough to meet needs, but in other cases, such as with sugar pine or western white pine and WPBR, and green ash and EAB resistance, one or more additional generations of breeding will be required. BBD does not reduce the time required for the first breeding cycle but can significantly reduce the number of breeding cycles to delivery time [49]. In theory, no characterization of gene function or resistance mechanisms are required at all during the discovery process. The

genomic patterns of polymorphisms in the best individuals are tested for predictive value in other phenotyped individuals, a process known as genomic selection (GS).

GS is a top-down strategy that can accurately predict QR in the presence of MGR and minor QTLs [50]. When genotypes with an aggregate of genes associated with a quantitative phenotype are identified, then the bottom-up process of targeted genotyping can, in theory, significantly reduce the amount of both phenotyping and genotyping necessary and may even allow for skipping phenotyping altogether in some breeding cycles. This will allow a cycle to proceed as fast as the generation time of the species in the QR breeding program. In the most ideal circumstances, targeted genotyping may permit the identification of superior parents directly from germplasm collections or natural populations, and this may be the most significant advance for many tree species threatened by new pathogens and pests. Much more effort is needed to facilitate this step, which would greatly assist newly initiated resistance programs. However, the four factors remain as limitations for the development and implementation of GS. We concur with the GS perspective shared by others [49], but also emphasize that, for resistance programs (and especially new resistance programs), perhaps the most feasible development would be verified genetic markers at least for MGR and large-effect QTLs. The use of genetic markers with verified associations could greatly increase the effectiveness of phenotypic selection in wild stands, an approach that would both expedite progress and reduce cost (due to reduction in the number of trees to test). However, at least in the near future or with the limited resources available, these are likely to be limited to MGR. Success using GS requires computational approaches appropriate for the sampling design, a sound understanding of the statistical power of the design chosen, and a phenotyping method predictive of field performance. Resistance work continues in many of the North American five-needle pine species, and traditional tree breeding methods will continue to be the focus. In the next 10 to 20 years, genomic resources may assist with additional markers to confirm parent identities and perhaps MAS for selecting new MGR parents in the field or determining whether more than one MGR is present in a species. The much more complicated nature of QR will likely limit the role of genomics in helping increase efficiency in forest tree resistance programs. Although technologies such as gene editing may lead to advances in some crop species, the complexity of most tree species and other factors will limit its use in resistance programs such as those with five-needle pines [51].

### 2.6. The Potential Role of Omics and Biotechnology in Shortening the Generation Time

A major obstacle to resistance breeding programs is the prolonged period of juvenility in forest trees, which can be as long as several decades. The silencing of genes regulating reproductive development resulted in early flowering in several tree species, demonstrating the feasibility of genome editing for the acceleration of the genetic improvement programs by accelerating reproductive maturity [52]. CRISPR-Cas9-based genome editing technology has been reported in forest conifer species [53], but its application may be limited in the foreseeable future [51]. Fortunately, other techniques have been developed to shorten the generation time of some species, including hormone treatments, drought stress, and various light regimes. Hormonal treatments have been reported to promote the production of both pollen and seed cones in western hemlock [54,55] and various other members of the Pinaceae family [56]. Hormone treatment has been effectively integrated into operational resistance breeding programs, such as the application of gibberellic acid ($GA_3$) to Port-Orford-cedar trees, stimulating the production of pollen and seed cones in young trees, accelerating breeding cycles [57]. Grafting mature American chestnut (*Castanea dentata* (Marsh.) Borkh.) and American beech onto seedling rootstock results in earlier flowering times than seedlings, which can reduce the time to produce first-generation populations from select parents [48,58]. Growing American chestnut under high-intensity light can shorten generation times even more by triggering early flowering in seedlings less than one year old [59]. Top grafting eucalyptus seedlings selected using GS or other omics-related techniques to a mature eucalyptus rootstock can induce flowering in the top graft

in three months, shortening the generation time to a few years and thus maximizing the genetic gain per unit of time [60].

### 3. Conclusions

In trees, traditional breeding has been successful in producing resistant trees for plantation forestry. More recently, it has also been used for producing genetically diverse, resistant populations of trees to restore some species of high ecological importance. Technology to increase the efficiency of resistance breeding would be of great benefit. When a pest or pathogen threatens a tree species, there is an initial effort to document the existence of MGR and QR. However, even when resistance is documented and a seedling screening assay is well-developed, many, if not most, of the field selections tested will have little or no resistance. The development of genetic markers and associated technologies to increase the efficacy of field selections may be the single most important development in a resistance program, particularly for the many non-economic species. Marker development may be feasible for MAS for MGR, with sufficient resources, but unfortunately it will likely be of limited assistance in developing GS strategies for QR. Species of highest economic value with resistance breeding programs already underway (e.g., loblolly pine) still need to breed for higher levels, higher frequency, and higher diversity of resistance. Utilizing genomic information for genomic selection may be warranted and potentially economically feasible, but its complexity and need to implement across multiple populations of trees may limit its overall utility.

Two other papers in this Special Issue address challenges to the potential use of genomics tools in tree breeding [51,61] and much of that applies to resistance breeding. Genomic information can potentially be valuable in resistance breeding, but the limits of the technology for forest trees, and, notably, for the next 10 to 20 years, needs further examination. In the cases where genomics will be useful in the long term, a much more concerted dialogue between geneticists and tree breeders at the onset of addressing the problem is needed. In addition, to expand work to more species, limitations in funding, infrastructure, and the skilled personnel needed to implement an operational resistance breeding and restoration program need also to be addressed. Ultimately, for many tree species affected by biotic challenges, society will have to prioritize which species are of highest concern and allocate sufficient resources to complete the delivery and establishment of resistant populations. The publication of examples of the successful uses of genomic information in forest tree resistance breeding, when it occurs, will help transition genomics from basic research to being fully integrated into operational resistance breeding programs in other tree species when the benefits outweigh the costs.

**Author Contributions:** All authors participated in the conceptualization and writing. All authors have read and agreed to the published version of the manuscript.

**Funding:** This work was funded in part by the USDA Forest Service.

**Acknowledgments:** We thank Gancho Slavov, Glenn Howe, and Jaroslav Klapste, the Special Issue editors for their comments on an earlier version of this paper. We also thank the three reviewers for their comments. The findings and conclusions in this publication are those of the authors and should not be construed to represent any official USDA or U.S. Government determination of policy.

**Conflicts of Interest:** The authors declare no conflict of interest.

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
