# Peer review of "Will Genomic Information Facilitate Forest Tree Breeding for Disease and Pest Resistance?"

_forests, doi:10.3390/f14122382_

Round 1

Reviewer 1 Report

Comments and Suggestions for Authors

The question asked by the authors “Will Genomic Information Facilitate Forest Tree Breeding for  Disease and Pest Resistance?” is an important and relevant question regarding breeding of forest trees but this article needs to be improved and answer queries before being considered for publication.

- Formatting according to the journal "forests" required for instance Keywords: “biotechnology” is bold;

- Please mention different resistance breeding programs for trees around the world; either in the form of a paragraph or table. There is no information on resistance tree breeding programs in the review article.

- A table mentioning the pest(s) including insects, fungi, etc. of forest trees, especially pine would be better to give readers an overview of pests infecting forest trees.

- Add a figure showing different seed zones

- Whetten et al., 2023. Genomic tools in applied tree breeding programs: Factors to consider. Forests. 2023, 14:169, gave a comprehensive review of tree breeding programs and their constraints. Authors may explain how their article is different from Whetten et al., 2023.

- The references required throughout the article to validate the statements.

- Formatting and uniformity in the reference section required

Author Response

The question asked by the authors “Will Genomic Information Facilitate Forest Tree Breeding for  Disease and Pest Resistance?” is an important and relevant question regarding breeding of forest trees but this article needs to be improved and answer queries before being considered for publication.

- Formatting according to the journal "forests" required for instance Keywords: “biotechnology” is bold;  **we have changed biotechnology to biotechnology no longer in bold font)**

- Please mention different resistance breeding programs for trees around the world; either in the form of a paragraph or table. There is no information on resistance tree breeding programs in the review article.

**MOST OF THE SUCCESSFUL ONES ARE DISCUSSED IN SNIEZKO & KOCH 2017.(reference [5] in the introduction, this is sufficient for a perspective; in addition there are relatively few well-established, successful applied programs around the world, but increasing interest due to the success of the ones mentioned in Sniezko & Koch 2017, etc.**

- A table mentioning the pest(s) including insects, fungi, etc. of forest trees, especially pine would be better to give readers an overview of pests infecting forest trees.

 That is beyond the scope of a perspective.  However, a number of the pathogens and pests are referenced in some of the references we have, but we have added three references at the beginning (now [1] to [3] that will provide the reader the literature to go to get a fuller picture.

- Add a figure showing different seed zones 

**Seed Zones can vary by species, we’ve added a sentence and a key citation, which is sufficient and will provide the readers will access to abundant literature on that subject.

- Whetten et al., 2023. Genomic tools in applied tree breeding programs: Factors to consider. Forests. 2023, 14:169, gave a comprehensive review of tree breeding programs and their constraints. Authors may explain how their article is different from Whetten et al., 2023.

  **NO NEED – WE FOCUS ON FOREST HEALTH AND BEYOND COMMERICIAL SPECIES AND WE CITE/ACKNOWLEDGE WHETTEN ET AL – a paper that was being prepared as the same time.  In addition, we also specifically distinguish between major gene resistance and quantitative disease resistance.in potential use of genomics.

- The references required throughout the article to validate the statements

**We have added a number of new references.

- Formatting and uniformity in the reference section required

**We have revised the reference section to be more uniform.

Reviewer 2 Report

Comments and Suggestions for Authors

The manuscript entitled “Will Genomic Information Facilitate Forest Tree Breeding for Disease and Pest Resistance?” by Sniezko et al. reviews the limitations and potentials of genomics application in resistence breeding of forest trees. It’s a good revew with keen insight and understanding, properly structured and well written. This manuscirpt will benefit to tree and crop breeders.

Some monor comments below:

-133-134. Please provide a reference.

-337-338. “This will be not generally be feasible for most tree species ” sounds stammer.

-398. Delete the full stop after “time”.

Comments on the Quality of English Language

The manscript is well-written.

Author Response

The manuscript entitled “Will Genomic Information Facilitate Forest Tree Breeding for Disease and Pest Resistance?” by Sniezko et al. reviews the limitations and potentials of genomics application in resistence breeding of forest trees. It’s a good revew with keen insight and understanding, properly structured and well written. This manuscirpt will benefit to tree and crop breeders.

Some monor comments below:

-133-134. Please provide a reference.  **A reference has been added**

-337-338. “This will be not generally be feasible for most tree species ” sounds stammer. 

**We are not certain what the reviewer was asking, but we think our revision may cover it**

-398. Delete the full stop after “time”.  **DONE**

Reviewer 3 Report

Comments and Suggestions for Authors

Thank you for very interesting and topical article. It was easy to read and the experience on the field was evident.

The only comment: In page 6, row 261-262, it was stated: “… in Eurasia where there is generally a high level of durable resistance.” There was no reference for this.

Author Response

Thank you for very interesting and topical article. It was easy to read and the experience on the field was evident.

The only comment: In page 6, row 261-262, it was stated: “… in Eurasia where there is generally a high level of durable resistance.” There was no reference for this.

**We have changed the wording and added two references**

Round 2

Reviewer 1 Report

Comments and Suggestions for Authors

The revised version of the manuscript includes additional relevant citations that have significantly improved the work. The article can be considered for publishing in the journal.